# Human Umbilical Mesenchymal Stem Cell Xenografts Repair UV-Induced Photokeratitis in a Rat Model

**DOI:** 10.3390/biomedicines10051125

**Published:** 2022-05-12

**Authors:** Yu-Show Fu, Po-Ru Chen, Chang-Ching Yeh, Jian-Yu Pan, Wen-Chuan Kuo, Kuang-Wen Tseng

**Affiliations:** 1Department of Anatomy and Cell Biology, School of Medicine, National Yang Ming Chiao Tung University, Taipei 112, Taiwan; ysfu@nycu.edu.tw; 2Institute of Anatomy and Cell Biology, School of Medicine, National Yang Ming Chiao Tung University, Taipei 112, Taiwan; tim2247749@gmail.com; 3Department of Obstetrics and Gynecology, Taipei Veterans General Hospital, Taipei 112, Taiwan; ccyeh39@gmail.com; 4Department of Obstetrics and Gynecology, National Yang Ming Chiao Tung University, Taipei 112, Taiwan; 5Department of Nurse-Midwifery and Women Health, National Taipei University of Nursing and Health Sciences, Taipei 112, Taiwan; 6Institute of Biophotonics, National Yang Ming Chiao Tung University, Taipei 112, Taiwan; simonpan62317@gmail.com; 7Department of Medicine, Mackay Medical College, New Taipei 252, Taiwan

**Keywords:** cornea, photokeratitis, umbilical mesenchymal stem cells, transplantation

## Abstract

Most patients with a corneal injury are administered anti-inflammatory medications and antibiotics, but no other treatments are currently available. Thus, the corneal injury healing is unsatisfactory, affects the vision, and has a risk of blindness in severe cases. Human umbilical mesenchymal stem cells exhibit pluripotent and anti-inflammatory properties and do not cause immunological rejection in the host. Rats were irradiated with type B ultraviolet (UVB) light to generate a stable animal model of photokeratitis. After irradiation-induced photokeratitis, human umbilical mesenchymal stem cells were implanted into the subconjunctival space of the lateral sclera, and the changes in the corneal pathology were evaluated. Three weeks after implantation, many mesenchymal stem cells were visible in the subconjunctival space. These mesenchymal stem cells effectively reduced the extent of injury to the adjacent corneal tissue. They accelerated the epithelial layer repair, reduced the inflammatory response and neovascularization, and improved the disorganization of collagen and fibronectin in the corneal stroma caused by the injury. In conclusion, xenografted human umbilical mesenchymal stem cells can survive in rat eye tissues for a long time, effectively support the structural integrity of injured corneal tissues, restore corneal permeability, and reduce abnormal neovascularization. This study provides a new approach to the treatment of photokeratitis.

## 1. Introduction

The cornea is the initial barrier to light entering the eye. There is a unique mechanism in the normal cornea to maintain a microenvironment in which corneal blood vessels are not neovascularized; thus, there are no blood vessels or blood cells. Any change may have deleterious effects, even a loss of vision in severe cases [1]. Although blindness is not life-threatening, it can significantly affect the quality of life. Changes include infection, chemical burns, physical injuries, or corneal transplants that can induce an inflammatory response and disrupt the homeostasis of the microenvironment of the cornea, promoting angiogenesis, the end result of a severe corneal injury [2]. When the cornea is injured, the vascular permeability of the corneal limbus increases, enabling inflammatory cells, such as neutrophils and macrophages, to enter the cornea [3,4]. Macrophages release proangiogenic factors and basic fibroblast growth factors (big), which promote the proliferation of vascular endothelial cells and play an important role in angiogenesis after severe corneal injury [5]. In addition, injured epithelial cells also release proangiogenic factors into the stroma. They induce keratocytes to become myofibroblasts, develop scabs, and secrete enzymes to break down the stroma and facilitate angiogenesis [6]. The inflammatory response and angiogenesis interaction and the cellular sources involved are diverse. For example, infiltrating macrophages, injured corneal endothelial cells, sprouting vascular endothelial cells, and keratocytes stimulated by injury secrete vascular endothelial growth factor (VEGF) proteins, resulting in abnormal corneal angiogenesis and affecting normal vision.

There are three types of ultraviolet rays: UVA (longer wavelength with less energy), UVB, and UVC (shorter wavelength with more energy). When sunlight reaches the atmosphere, UVC is completely absorbed (there is no UVC that appears on the Earth’s surface), leaving only 95% of UVA and 5% of UVB rays that reach the Earth’s surface. Although the percentage of UVB radiation is relatively small, the cornea is vulnerable to injury caused by UV light because of its high energy. Nearly 90% of the UVB rays reaching the eye are absorbed by the cornea [7,8,9,10]. A UV light-induced corneal injury is called photokeratitis (also known as snow blindness) and is an acute injury that usually shows symptoms beginning 6 h after exposure. The symptoms include injury and shedding of the epithelial cells, resulting in severe foreign body sensation, photophobia, and watery eyes. If the condition is severe, it can lead to blurred (fuzzy) vision and corneal edema [11]. In addition to photokeratitis, pterygium may also occur from chronic cornea injury by UV radiation [12].

When the corneal injury is too severe, surgery is the only solution; thus, researchers are now actively engaged in evaluating cell therapy as a treatment strategy. Human umbilical mesenchymal stem cells also exhibit pluripotent properties. They can express genes from each germ layer during development and differentiate into specific cells derived from each germ layer [13], such as neurocytes in the ectoderm, chondrocytes in the mesoderm, and hepatic cells in the endoderm [14]. Human umbilical mesenchymal stem cells also exhibit immunomodulatory activity. A low expression of major histocompatibility complex class II renders them less likely to produce immunological rejection reactions [15]. They regulate the immune system by suppressing T cells, B cells, natural killer (NK) cells, and dendritic cells [16].

Our previous studies demonstrated that human umbilical mesenchymal stem cells (HUMSCs) effectively treat spinocerebellar ataxia, epilepsy, osteoporosis, liver fibrosis, peritoneal fibrosis, and pulmonary fibrosis [17,18,19,20,21,22,23]. These attributes make human umbilical mesenchymal stem cells a good source for xenografting. In this study, we used UV light to induce corneal injuries in rats to track the pathological changes in the cornea and injected human umbilical mesenchymal stem cells into the subconjunctival space to determine whether a therapeutic effect was achieved. Here, we conducted an in vivo longitudinal observation of the structural changes of the cornea using a multifunction optical coherence tomography (OCT). OCT provides real-time, two-dimensional (2D), and three-dimensional (3D) images with high resolution and adequate depths without a contrast agent or radiation. Therefore, OCT has become a useful tool for clinical ophthalmology. Expanding conventional OCT to multifunctional OCT enables the simultaneous acquisition of the corneal microstructure, blood vessel morphology, and birefringence structures. Changes in the above parameters may indicate the presence of tissues in the early stages of a pathological process [24,25].

## 2. Materials and Methods

The Research Ethics Committee of Taipei Veterans General Hospital (202112007CC) and the Animal Research Committee of MacKay Medical College (IACUC-A1110015) approved using human umbilical cords and laboratory animals.

### 2.1. Isolation and Culture of Human Umbilical Mesenchymal Stem Cells

The umbilical cords were collected aseptically and stored in Hank’s balanced salt solution buffer at 4 °C. The mesenchymal tissue in Wharton’s jelly was then diced into cubes 0.5 cm on each side and centrifuged at 250× *g* for 5 min. After removal of the supernatant fraction, the precipitate that contained mesenchymal tissue was washed with serum-free Dulbecco’s modified Eagle’s medium (DMEM) and centrifuged at 250× *g* for 5 min. After aspiration of the supernatant fraction, the mesenchymal tissue in the precipitate was treated with 0.0125% type I collagenase solution (Sigma-Aldrich, St. Louis, MO, USA) at 37 °C for 18 h, washed, and further digested with 0.025% trypsin at 37 °C for 30 min. The treated umbilical tissue then was cultured until the HUMSC migration and proliferation. Finally, the HUMSCs were stored in liquid nitrogen for later transplantation. The HUMSCs were collected between the 10th and 15th passages for transplantation into rats in this study.

### 2.2. Experimental Animals

Three-week-old male Sprague–Dawley rats were obtained from the Experimental Animal Center and raised to an age of seven to eight weeks, weighing approximately 270–300 g. The animals were exposed to 12 h of light per day (7:30 a.m. to 7:30 p.m.) at a constant air-conditioned temperature (22 ± 2 °C) and were provided adequate feed and drinking water.

### 2.3. Experimental Grouping

The rats were divided into three groups: Normal group (*n* = 13), UV group (*n* = 32), and UV + HUMSCs group (*n* = 13). The rats were anesthetized with zoletil 50 and xylazine hydrochloride (Sigma-Aldrich) for the UV group by intraperitoneal injection. Left eyes of the experimental animals were placed on a table facing upward, and the light-induced corneal injury experiment was conducted for five consecutive days using a lamp (model UVM-18, P/N 180063-01, UVP Inc., San Gabriel, CA, USA) with a power of 8 watts, a UV wavelength of 302 nm, and an energy of 550 µW/cm^2^ per day. For the UV + HUMSCs group, the lower eyelid of the rat was propped open after five consecutive days of UV light-induced corneal injury, and 2 × 10^6^ HUMSCs were implanted into the subconjunctival space of the lateral sclera using a Hamilton syringe (Figure 1A). 

### 2.4. Intraocular Pressure Measurement

The intraocular pressure of the rats was measured using an iCare Lab tonometer (TV02) from 10:00 a.m. to 11:00 a.m. Six consecutive measurements were obtained, and the average value was considered the daily intraocular pressure.

### 2.5. Corneal Surface Injury Evaluation

The drug fluorescein sodium salt (Sigma F6377; 460–515 nm) was used at a concentration of 2.5% (0.625 g/25 mL saline dissolved in saline) and stored at 4 °C. The fluorescein sodium solution was applied to the corneal surface of the rats using a 200 μL micropipette (the rats blinked three times with the assistance of human hands before applying the drops to avoid the effect of corneal dryness in the results). Photos were taken for observation after the excess fluorescein sodium solution was absorbed with Kimwipes. The corneas were washed with physiological saline after the experiment. The green fluorescence of corneal fluorescein was quantitated using ImageJ software to square out the entire corneal area. The ratio of the green fluorescence area to the entire corneal area was calculated.

### 2.6. Histology and Immunohistochemistry

The animals were anesthetized with zoletil 50 and xylazine hydrochloride (Sigma 23076359) via intraperitoneal injection and then perfused. Tissues were first perfused with 0.9% physiological saline, then fixed by perfusion with 4% paraformaldehyde (Sigma 10060) and a fixative solution consisting of 7.5% picric acid (Sigma 925-40) dissolved in 0.1 M PB, followed by embedding.

The ocular tissues were sliced to a thickness of 7 μm. The dried eye tissue sections were dewaxed, hydrated, and placed into a 0.296% trisodium citrate dihydrate solution. After the solution containing the tissue section was heated to boil, it was incubated for 10 min, then cooled until reaching room temperature. The sections were then immersed in 0.1 M PBS solution for 30 min to complete the recovery of the tissue antigens. Next, immunostaining was performed. The sections were stained with hematoxylin–eosin, and intracorneal infiltrating macrophages were stained by anti-ED-1 (Millipore, Burlington, MA, USA) immunostaining. Keratocytes were stained with anti-ALDH3A1 (ProteinTech Group, Chicago, IL, USA) to assess the keratocyte distribution in the cornea and identify the distribution of the myofibroblasts associated with fibrosis in the cornea. The vessels were stained with anti-CD31 (Abcam, Cambridge, MA, USA). The specific cytoskeleton of the corneal epithelial cells was stained with anti-Keratin-12. The arrangement of the extracellular matrix was stained with anti-fibronectin immunostaining to identify the changes in collagen I in the cornea. Collagen I was detected by anti-collagen I immunostaining, and collagen III was stained by anti-collagen III immunostaining. In addition, the distribution of xenografted HUMSCs was determined by immunostaining with anti-human nuclei antigen.

### 2.7. Visualization of DiR-Labeled HUMSCs

HUMSCs were treated with 1,1′-dioctadecyl-3,3,3′,3′-tetramethylindotricarbocyanine iodide (DiR) for 30 min before transplanting into the subconjunctival space of the UV + HUMSC group. With an excitation wavelength of 748 nm, the DiR-labeled HUMSCs emitted near-infrared fluorescence at 780 nm, which was used to detect the survival of the HUMSCs in vivo using an optical imaging system (Biospace Lab Optima, Des Moines, IA, USA).

### 2.8. Multifunction OCT

This study used an available multifunction OCT system in W. C. Kuo’s lab [24,25]. The light source was a swept-source laser with a central wavelength of 1310 nm, spectral bandwidth of 100 nm, and sweeping speed of 100 kHz. The axial resolution of the imaging system was measured at 11 mm, whereas the lateral resolution was approximately 7 mm. After data acquisition, several postprocessing steps were performed to achieve a multi-contrast image display (including thickness map, light scattering map, birefringence map, and OCT angiography map), as shown in Appendix A.

### 2.9. Statistical Analysis

All experimental data were expressed as the mean ± SEM (mean standard error). A one-way ANOVA or two-way ANOVA made comparisons between means, and multiple comparisons were made using Fisher’s least significant difference test. A *p*-value < 0.05 was considered statistically significant.

## 3. Results

### 3.1. Ultraviolet Light Exposure Causing Corneal Injury and Corneal Opacity

On the first day of UV irradiation, there was no significant difference between the appearance of the cornea in the UV group compared with the Normal group, in which clear pupils were observed in both. On the third day of UV irradiation, the corneal transparency of the UV group began to decrease and exhibited opacity. Corneal transparency did not recover on the 21st day after irradiation, and there were blisters on the corneal surface and abnormal neovascularization in the center of the cornea. In the UV + HUMSC implantation group, the corneas exhibited the same opacity as the UV group initially; however, on the 14th day after the end of UV irradiation, the corneas of the experimental animals in the UV + HUMSC group were clearer compared with those of the UV group. Thus, the pupil structure could be observed through the cornea, and there were no blisters on the corneal surface and less abnormal neovascularization in the center of the cornea. The results indicated that xenografting with HUMSCs reduced the pathology of the damaged corneas (Figure 1B).

### 3.2. Histological Staining to Observe Changes in the Morphology of Tissue Sections

The histological sections of the central region of the cornea in the Normal group exhibited a regular arrangement of the corneal epithelial cells. They had a uniform, regular, and compact arrangement of the extracellular matrix of the stroma, with only a small amount of keratocytes after H&E staining. On the third day of UV irradiation, most of the corneal epithelial cells were dead, and this condition continued until day 7 after the end of UV irradiation. On day 21 of irradiation, gradual regeneration of the corneal epithelial cells was visible; however, the cells appeared as a pseudo-keratinized epithelium with a distinctly different structure from that of the normal cornea. On day 5 of irradiation, many cells began to infiltrate the corneal stroma, and the thickness increased significantly. Although the corneal thickness gradually recovered on day 5 after UV irradiation, many cells accumulated in the stroma with a disorganized arrangement of the extracellular matrix. The H&E-stained corneal tissue sections revealed a corneal thickness in the UV group, which gradually increased over time, from day 5 of irradiation to day 3 after the end of irradiation, and the corneal thickness was 169.3 ± 34.7 μm and 163.3 ± 6.1 μm, respectively. These values were significantly higher than that of the Normal group (*p* < 0.05). The thickness gradually decreased on day 5 after UV irradiation. Eventually, there was no statistical difference compared with the corneal thickness in the Normal group, indicating a transient increase in the thickness of the injured corneas (Figure 2A,D).

The repair of the epidermal cells occurred in the UV + HUMSC group on day 21 after UV irradiation. The new epithelial layer appeared similar to the epithelial morphology of the Normal group, and there was less neovascularization in the stroma. In addition, there was no statistical difference in the thickness of the central cornea compared with that of the Normal group. The histomorphology observation revealed that the degree of corneal pathology was improved by the HUMSC xenografts (Figure 2C,D).

### 3.3. Observation of Changes in Pathological Corneal Thickness in In Vivo Images Using OCT

Figure 2E and Appendix A show the enface thickness colormap, in which dashed lines illustrate the 2D OCT structure. The corneal thickness map of the Normal group exhibited a uniform blue color at each observation time, indicating that the corneal thickness did not change over time. However, when irradiated with UV light, the cornea thickness increased sharply (from the periphery) until the 8th day of the experiment (D3). It slowly decreased from the center and, finally, approached the initial corneal thickness. Similar to the injury group, the corneal thickness in the treatment group increased, but the increase was much smaller. 

We selected a circle with a 1-mm radius around the cornea center in Figure 2E to calculate the average thickness. The average corneal thickness in the central region of the quantitative OCT images indicated that the corneal thickness in the Normal group was maintained at 123.2 ± 12.1 μm and −140.9 ± 10.9 μm. There was no significant difference between corneal tissue thickness in the UV group on the first day of irradiation compared with the Normal group. However, on day 5 of the UV irradiation and day 3 after the end of the UV irradiation experiment, the corneal thickness gradually thickened because of edema. The corneal thickness was 516.8 ± 242.2 μm, 406.2 ± 183.7 μm, and 348.9 ± 48.9 μm at the end of the 3rd, 5th, and 7th days after UV irradiation exposure, respectively, which was statistically different from that of the Normal group (*p* < 0.05). This suggests that UV irradiation caused corneal inflammation, swelling, and thickening. The degree of corneal edema gradually decreased between days 14 and 21 after irradiation. There was no statistical difference between the thickness measured on day 21 after irradiation compared with that of the Normal group (Figure 2F). In the UV + HUMSC group, the corneal thickness was 353.8 ± 173.1 μm and 299.9 ± 126.7 μm on days 3 and 5 after irradiation, respectively, showing a statistically significant difference in thickening compared with that of the Normal group (*p* < 0.05). On day 7 following irradiation, the corneal thickness was approximately 275.6 ± 113.8 μm, with no statistical difference compared with that of the Normal group. This indicated that HUMSC xenografts reduced the pathological thickening of the injured corneas and shortened the duration of the corneal thickening and edema (Figure 2F).

### 3.4. HUMSC Xenografting Repairs UV-Induced Corneal Epithelial Injury

A fluorescein stain was applied to the eye surfaces of living rats to determine the corneal surface integrity. The stained area was used to quantify the area of injury. The epidermal cell layer is intact in normal corneas, and the fluorescence dye does not remain on the corneal surface. However, if the cornea is injured, the epidermal cell layer is incomplete, and the dye remains on the surface and emits green fluorescence after stimulation with blue light. The green area represents the percent integrity of the corneal surface. The corneas of the rats in the Normal group were intact, and no dye was retained in the corneas throughout the experiment. In the UV group, however, a large area of green fluorescence was retained on the surface from days 3 to 5 of exposure, presumably because of a large amount of corneal epithelial cell injury, resulting in an incomplete epithelial cell layer. The green fluorescence on the corneal surface remained until day 21 after irradiation; however, the area reduced significantly, presumably because the injured corneal epithelial cells were gradually repaired. In the UV + HUMSC group, the retained area of the dye on the eye surface was the same as that of the UV group during the irradiation period; however, no fluorescent dye was on days 14 and 21 after irradiation observed on the eye surface. There was a statistically significant decrease compared with the UV group, suggesting that HUMSCs accelerate the repair and growth of epidermal cells (Figure 3A,B).

The corneal tissue sections were stained with H&E to observe the corneal epidermal cells in the UV group. The epidermal cell layer disappeared on day 3 of the UV irradiation, which continued until day 7 after the UV irradiation. Although a dark-stained cell layer reappeared on day 21 after UV irradiation, its morphology differed from that of the normal corneal epidermal cell layer. In contrast, the UV + HUMSC group exhibited a corneal epithelial layer similar to the Normal group on day 21 after irradiation (Figure 3C). The cytoskeleton of the corneal epithelial cells was labeled with anti-keratin-12 antibodies to observe the cytoskeletal proteins within the corneal epithelial cells. In the corneas of the Normal group, keratin-12 was only distributed in the epidermal cell layer. In the UV group, keratin-12 was completely undetectable in the epidermal cell layer on day 5 of irradiation, which was maintained until day 21 after irradiation. However, in the UV + HUMSC group, keratin-12 was already present in the epidermal cell layer on day 21 after irradiation, indicating that the HUMSC xenografts induced epidermal cell renewal with a normal expression of cytoskeletal keratin-12 (Figure 3C,D).

### 3.5. Effect of HUMSCs Xenografting on the Arrangement of Extracellular Matrix Proteins, Fibronectin, Collagen I, and Collagen III, in the Structure of the Corneal Stroma

Figure 4A,C (upper row) show the birefringence enface map for the Normal, UV, and UV + HUMSC groups, respectively. The dashed lines illustrate 2D phase retardation images (see lower row). The birefringence map of the Normal group (Figure 4A) showed a green to yellow color at each observation time point, indicating that a healthy cornea exhibits an intrinsic birefringence property because of the regular arrangement of the collagen layers. Upon exposure to UV light (Figure 4B), the birefringence decreased sharply from d3 and, finally, approached near-zero retardation (i.e., blue color). UV damage may lead to the destruction of the structure of the cornea, whereas the extracellular matrix secreted by fibroblasts also causes the internal structure to become chaotic; thus, the birefringence property disappears. Similar to the injury group, the birefringence in the treatment group (Figure 4C) is still decreased. However, the birefringence map appears as a green to yellow color again on days 14 and 21 of the treatment group, indicating that the cornea’s internal structure arrangement (after treatment with HUMSC) is more similar to a healthy cornea than the injured group.

In the normal group, fibronectin was evenly distributed and regularly arranged in the corneal stroma in the Normal group but not in large amounts. Fibronectin was present in large amounts in the corneal stroma on day 21 following UV irradiation of the UV group; however, the protein arrangement was disorganized. In the UV + HUMSC group, there was a significant amount of fibronectin in the corneal stroma, with some areas arranged more regularly. This indicates that HUMSC xenografting accelerated the expression and reorganization of fibronectin in the injured corneas (Figure 4D).

The Normal group exhibited a regular arrangement of collagen I within the corneal stroma. In the rats in the UV-treated group, there was a large but uneven distribution of collagen I in the corneal stroma with a disorganized arrangement. In the UV + HUMSC group, collagen I in the corneal stroma was disorganized in some areas but was closer to the Normal group in other areas. In contrast, collagen I exhibited a regular arrangement (Figure 4E).

The anti-collagen III antibody was used to identify collagen III in the cornea, and the results indicated that the corneas of the Normal group contained small amounts of collagen III, which was mostly present in the epidermal cell layer, with minimal expression in the stroma. In the UV group, no collagen III was observed in the epidermal layer of the cornea on day 21 following irradiation, and collagen III was minimally expressed in the stroma. In the UV + HUMSC group, collagen III was observed in the epidermal layer of the cornea but was not detected in the stroma, which was similar to the distribution in the Normal group (Figure 4F).

### 3.6. Effect of HUMSCs Xenografting on Immune Cell Infiltration in the Cornea

Figure 5A shows the light scattering map. Transparent normal corneal tissue exhibits very weak light scattering. Following UV damage, the light scattering area (i.e., green area) is largely increased. When the cornea was treated with stem cells, the time point for the increase in light scattering occurred later than that in the injury group, and the increase in light scattering was also reduced. A statistical analysis (Figure 5B) revealed that, on the 12th (D7) and 19th (D14) and the 26th day (D21), there were significant differences in the scattering area between the treatment group and the injury group, which suggests that adding stem cells to the cornea can reduce the increase in the light scattering area.

Macrophages in the cornea were labeled with ED-1 antibodies to examine the inflammatory response in the central corneal region. The results indicated that few macrophages were observed in the Normal group. The UV group contained many macrophages infiltrating the corneal stroma in the central region and some near the blood vessels. The UV + HUMSC group showed significantly fewer macrophages in the corneal stroma. We speculated that HUMSC xenografting results in less inflammatory cell infiltration during chronic inflammation, thus reducing the pathology caused by inflammation (Figure 5C,D).

### 3.7. OCT Angiography (OCTA) Shows That HUMSCs Xenografting Reduces Abnormal Angiogenesis in the Cornea

Blood vessels were present at the limbus of the junction between the cornea and the sclera, but no blood vessels were observed in the corneal region of the UV group. On day 3 after irradiation, blood vessels began to sprout and extend toward the injured cornea. As time passed, the lengths of the extended blood vessels became longer, and on day 14, after UV irradiation, the lengths of the blood vessels reached the center of the cornea. The blood vessels almost extended over the entire cornea, and the extension of the blood vessels also reached saturation at this time. In the UV + HUMSC group, although there was also angiogenesis in the neovascularization, the amount was lower by days 14 and 21 after irradiation. The proportion of vessels to the total area of the cornea was quantified. The results showed a statistically significant increase in angiogenesis in both the UV group and the UV + HUMSC group compared with that of the Normal group at days 3, 5, 7, 14, and 21 after UV irradiation (*p* < 0.05). However, the UV + HUMSC group exhibited a significant decrease in the amount of neovascularization compared with that of the UV group (*p* < 0.05), indicating that HUMSC xenografting reduces angiogenesis in injured corneas (Figure 6A,B).

The distribution and change in the blood vessels of the cornea were determined using CD31 antibodies. Blood vessels were not observed in the cornea at the central region of the eye surface in rats from the Normal group. In the UV group, many blood vessels were evident in the cornea’s stroma. In the UV + HUMSC group, a small number of blood vessels was detected in the corneal stroma (Figure 6C).

### 3.8. HUMSCs Xenografting Promotes the Expression of Aldehyde Dehydrogenase 3A1 (ALDH3A1)

The distribution and changes of ALDH3A1 in the cornea were determined using ALDH3A1 antibodies. ALDH3A1 was present in fan-shaped keratocytes and in the extracellular matrix of the Normal group, and ALDH3A1 was also expressed in the epidermal cell layer. In contrast, no ALDH3A1-stained keratocytes were observed in the corneal stroma on day 21 after UV irradiation in the UV group, although ALDH3A1 was expressed in the epidermal cell layer. In the UV + HUMSC group, ALDH3A1 staining was observed not only in the epidermal cell layer but also in the stroma in fan-shaped keratocytes, similar to that of the normal corneas. This indicates that HUMSC xenografting maintains normal keratocyte activity, which, in turn, helps to repair the injured cornea (Figure 7A).

### 3.9. HUMSCs Survive in Rat Cornea after Transplantation

After the transplantation of high-dose DiR-labeled HUMSCs, red signals appeared in the rat eye on day 21 (Figure 7B). Immunohistochemical staining on samples of the cornea of the UV + HUMSC group on day 21 indicated that numerous HUMSCs survived at multiple locations near the limbus (Figure 7C(C1,C2)); however, a few HUMSCs were found in the stroma of the cornea (Figure 7C(C3).

## 4. Discussion

It is indicated that the use of UV light to injure the cornea causes comprehensive damage and affects the entire cornea rather than being limited to the epithelial and stromal layers. A study demonstrated UVB irradiating rat eyes continuously for five days, three minutes per day, and observed the pathological changes after complete injury. This injured the epithelium and changed the thickness and structure of the entire cornea [10]. However, we hoped to extend the pathology to three weeks after injury to more clearly determine the effectiveness of the treatment. In previous studies, the main treatment modalities for corneal cell treatment could be divided into two major categories: conjunctival injection or the implantation of cells into the amniotic membrane or collagen gel, which may be applied directly to the corneal surface. In the experiment, treatment was administered for one week after injury using a rat model of corneal alkali burn. The differences between the two bone marrow mesenchymal stem cell delivery methods were compared, including the subconjunctival injection or cell implantation in the amniotic membrane, followed by application to the cornea [26]. The results indicated that, although both methods exhibited therapeutic effects, the epidermal repair of the cornea by fluorescein injection was faster compared with that of the amniotic membrane implantation. There was also less angiogenesis and secretion of the VEGF proangiogenic factor. The number of cells that could be delivered using a dressing was fewer compared with a direct conjunctival injection because of the dressing material. Therefore, we selected subconjunctival injection to deliver HUMSCs in the present study. Grazarya et al. observed the pathology for four weeks after injury, longer than our experiment. However, upon observing the visual appearance of the eyeballs in the injury group, the corneas were not cloudy, the pupils were visible, and angiogenesis was sparse compared with our experiment. UV light affected the entire cornea, which became cloudy and opaque. Meanwhile, in terms of changes in the corneal stroma, the stroma in their injury group was more regularly arranged, without many infiltrating cells. Therefore, our experiment better simulates the therapeutic effect that stem cells play during severe corneal injury.

Mesenchymal stem cells have a wide range of sources and can be obtained from a variety of sources, such as bone marrow, adipose tissue, and fetus umbilical cord tissue [27,28,29]. In a previous study, bone marrow mesenchymal stem cells were administered intravenously to mice after scraping the corneal epithelium. As a result, the retention area of the corneal surface dye fluorescein was significantly reduced, and keratin-12 expression in the corneal epidermis increased after 3 days [30]. In the present study, HUMSC xenografting into the subconjunctival space after UV light exposure to the corneal tissue was also effective in repairing the injured corneal surface. There was also a significant reduction in fluorescein staining on the corneal surface. This was an indication that the epithelial cells were more structurally intact, and a new epithelial cell layer was observed for the cell types in the section. The cytoskeletal protein keratin-12, which is characteristic of normal corneal epithelial cells, was also observed on the cornea’s surface by immunohistochemical staining.

Another study used NaOH drops on the cornea to cause corneal alkali injury and then used nanofiber scaffolds to apply bone marrow mesenchymal stem cells to the corneal surface. A significant reduction of neovascularization in the cornea was observed by immunostaining for VEGF [31]. Recent studies have indicated that regulating the phenotype and recruitment of macrophages can encourage the functional improvement of ocular inflammation and remodeling [32,33]. In the present study, instead of using nanofiber scaffolds, HUMSCs were directly injected into the subconjunctival space, and a similar therapeutic effect was observed. In addition, immunostaining with anti-αSMA revealed no significant vascularization in the central cornea. In addition, the results of immunostaining with anti-ED-1 also showed less infiltration of the inflammatory macrophages in the central cornea.

After physically injuring rat corneas, another study applied a fibrin gel containing corneal marginal mesenchymal stem cells to the corneal surface. After treatment, multifunction OCT was used to achieve a multi-contrast image display (including a thickness map, light scattering map, birefringence map, and OCTA map), followed by a quantitative analysis. The photorefractive properties of the cornea were more similar to that of the normal group, the morphology of the newly synthesized collagen was relatively normal, and the stem cells produced type 1 collagen in the normal corneal stroma [34]. Xenografting HUMSCs also helped to restore the corneal structure, which recovered the birefringent characteristics of the normal cornea. The phenomenon of birefringence is observed in media containing ordered arrays of anisotropic structures, such as in the cornea. The loss of birefringence in the cornea may indicate changes in the tissue functionality, structure, or viability after damage [35]. Furthermore, the immunostaining results for anti-fibronectin, anti-collagen I, and anti-collagen III revealed that the extracellular matrix arrangement and collagen type in the injured cornea was maintained similar to that of the normal corneal structure.

Previous in vivo studies used a corneal alkali burn and subconjunctival injection of bone marrow mesenchymal stem cells [36]. The results indicated that the number of inflammatory cells in the stroma decreased, the number of inflammatory factors secreted decreased, and the mesenchymal stem cells did not migrate into the injured cornea. This indicated that the mesenchymal stem cells remained at the injection site and relied on paracrine action to inhibit the production of inflammatory factors from achieving a therapeutic effect. Compared with this experiment, we used anti-human nuclei antibodies but did not observe the presence of mesenchymal stem cells in the cornea, whereas a large number of HUMSCs were present in the bulbar conjunctival region. However, the extent of cell signaling, and the cytokine effects need to be further investigated. Finally, we anticipate that the results of this study will serve as a basis for clinical studies on optical eye injuries.

## 5. Conclusions

Xenografted human umbilical mesenchymal stem cells can survive in rat eye tissues for a long time, effectively supporting the structural integrity of injured corneal tissue. In vivo longitudinal studies of cornea pathogenesis and treatment outcomes were presented that utilized multifunction OCT-based approaches. Additionally, the histopathological analysis revealed a greater degree of the structural integrity of the epithelium of the corneal surface, complete preservation of the extracellular matrix type and arrangement in the cornea, and reduced neovascular growth in the cornea. These observations identified the xenografting of human umbilical mesenchymal stem cells as being an effective way of reducing the pathological time course while promoting the repair and regeneration of the injured cornea.

## Figures and Tables

**Figure 1 biomedicines-10-01125-f001:**
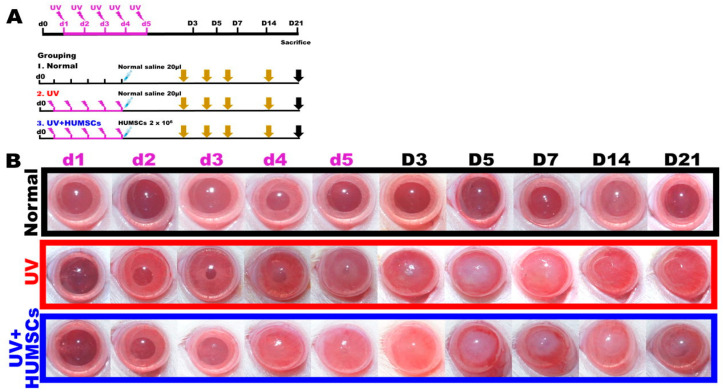
Experimental protocol and effects of UV irradiation and HUMSC treatment. The experimental flowchart for inducing photokeratitis, the transplantation of the HUMSCs, and the time course of the experiments in this study. The Normal group was administered a subconjunctival injection of 20 μL of normal saline on day 5. The daily UV was performed on day 1 through day 5. The experimental animals of the UV group served as UV controls exposed to UV light and were treated with subconjunctival injection of 20 μL of normal saline. The experimental animals of the UV + HUMSC group were exposed to UV irradiation and administered a subconjunctival injection of HUMSCs on day 5 (**A**). The corneas of the control rats were transparent and intact. UV radiation caused severe injury to the cornea, including apparent decomposed smoothness and opacity in the UV group. In contrast, significant enhancement of corneal organization and transparency were observed in the UV + HUMSC group compared with the corneal damage observed in the UV-treated animals on day 21 after UV irradiation (**B**).

**Figure 2 biomedicines-10-01125-f002:**
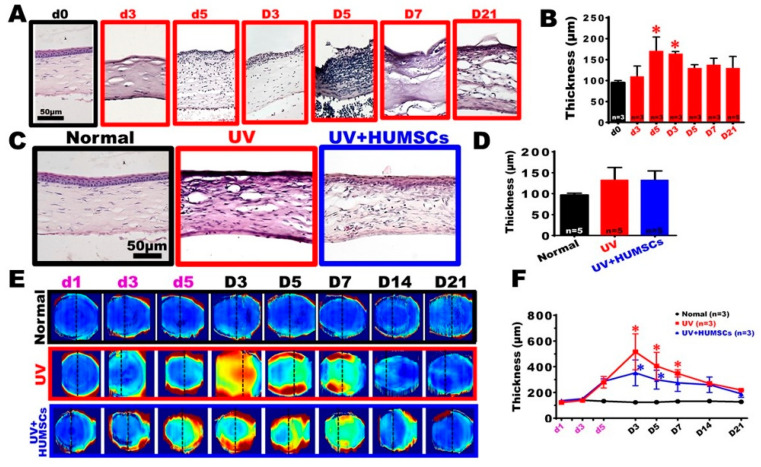
HUMSC transplantation alleviates corneal edema following UV irradiation. Hematoxylin staining and thickness analyses of the histological sections (**A**,**B**). Non-keratinized stratified squamous epithelium and mostly developed stroma were observed in the Normal group. Histological analysis of the irradiated cornea, swelling, and accumulation of infiltration cells in the stroma layer contributed to the thickness. The graph shows that the thickness significantly increased on day 5 and day 3 after UV irradiation (**B**). On day 21, after UV irradiation, pseudo-keratinization was observed in rats of the UV groups. In contrast, a significant enhancement of the intact epithelium was observed in the rats of the HUMSC groups (**C**). The graph shows the thickness of the corneas on day 21 after UV irradiation (**D**). The corneal thickness map (**E**) and average thickness within a 1-mm radius around the cornea center (**F**) were performed by quantitative analysis of the 3D OCT images. * vs. the Normal group at the same time, *p* < 0.05.

**Figure 3 biomedicines-10-01125-f003:**
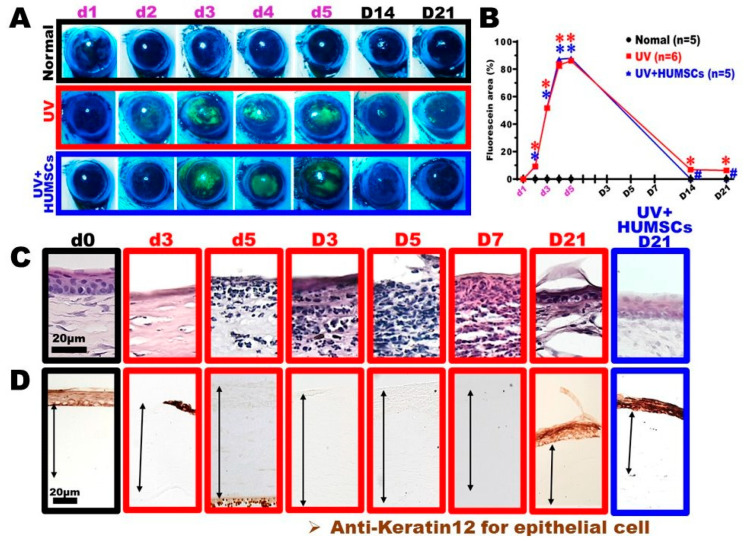
HUMSCs transplantation diminishes corneal epithelial disorganizations following UV irradiation. Corneal fluorescein staining and quantitative analyses of corneal integrity were conducted. The corneal surface was intact, and no staining of fluorescein was observed in the Normal group. UVB irradiation caused serious damage to the corneal tissues, including deteriorated smoothness and staining with fluorescein (**A**,**B**). Corneal sections were stained with hematoxylin (**C**) and anti-keratin-12 (**D**) and indicated the presence of the corneal epithelium. Anterior epithelial detachment and pseudo-keratinization were observed in the rats of the UV group. In contrast, a significant enhancement of the corneal epithelial organization was detected in the UV + HUMSC group rats. The animal number per group is shown in the figure. * vs. the Normal group simultaneously, *p* < 0.05; # vs. the UV group at the same time, *p* < 0.05.

**Figure 4 biomedicines-10-01125-f004:**
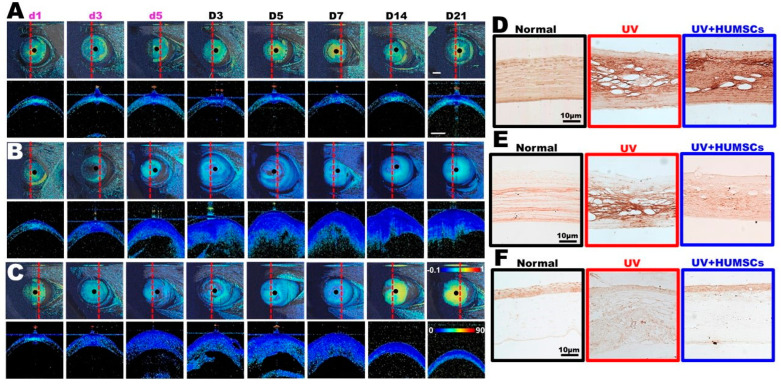
HUMSCs transplantation enhances the corneal collagen organization after UV irradiation. Multifunction OCT showing the birefringence map (upper row), in which dashed lines illustrate representative 2D phase retardation images (lower row) in the Normal (**A**), UV (**B**), and UV + HUMSC (**C**) groups. Immunohistostaining shows the distribution of fibronectin (**D**), collagen I (**E**), and collagen III (**F**) expressions in the stromal layer of corneas in the Normal, UV, and UV + HUMSC groups. The results show the distribution and arrangement of the extracellular matrix in the UV + HUMSC group were similar to those in the Normal group.

**Figure 5 biomedicines-10-01125-f005:**
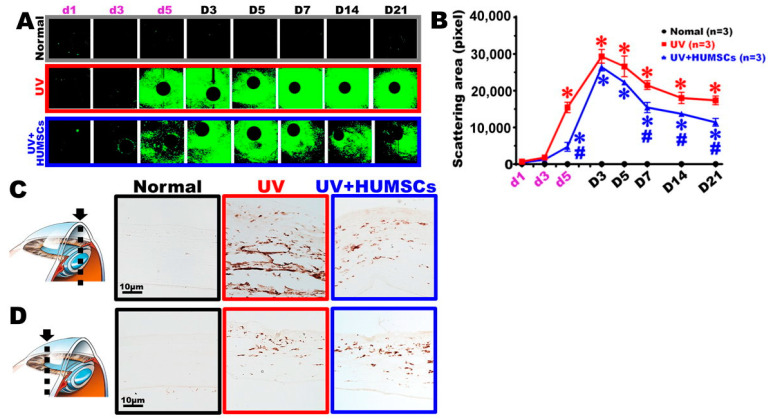
HUMSC transplantation reduces the number of macrophages in the cornea and decreases the light scattering. Multifunction OCT shows the light scattering enface maps (**A**) and the analysis of the scattering area (**B**). Adding stem cells to the cornea can reduce the increase in the light scattering area. Immunohistochemical staining of anti-ED1 was performed to detect activated macrophages in the central (**C**) and peripheral areas (**D**) of the corneas. Strong infiltration of the macrophages was present after UV irradiation, whereas limited activated macrophages were detected in the central area of the corneas in the UV + HUMSC group. The UV + HUMSC group showed significantly fewer macrophages in the corneal stroma. * vs. the Normal group simultaneously, *p* < 0.05; # vs. the UV group at the same time, *p* < 0.05.

**Figure 6 biomedicines-10-01125-f006:**
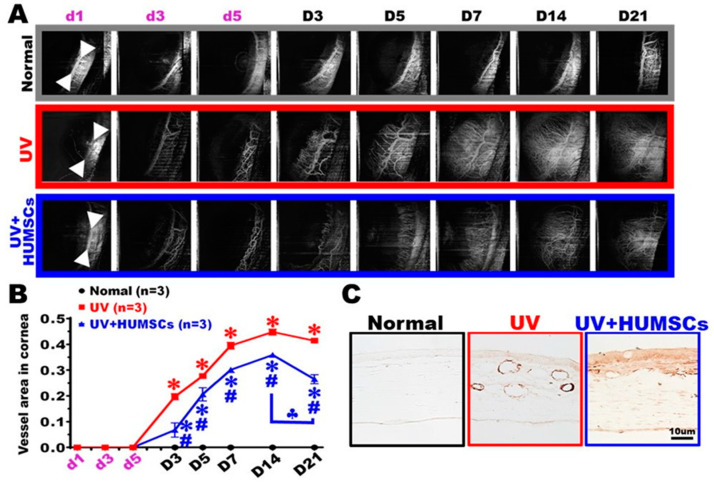
HUMSC transplantation diminishes angiogenesis after UV irradiation. Multifunction OCT shows the OCTA enface image (**A**) and analyzes the vessel area in the cornea (**B**). The vascular endothelial cells of the parietal peritoneum were labeled with a CD31 antibody. The results showed a large number of vessels in the stroma of the UV group. The number of blood vessels substantially decreased in the cornea of the UV + HUMSC group (**C**). * vs. the Normal group simultaneously, *p* < 0.05; # vs. the UV group at the same time, *p* < 0.05; ♣ vs. the UV + HUMSC group at day 14, *p* < 0.05.

**Figure 7 biomedicines-10-01125-f007:**
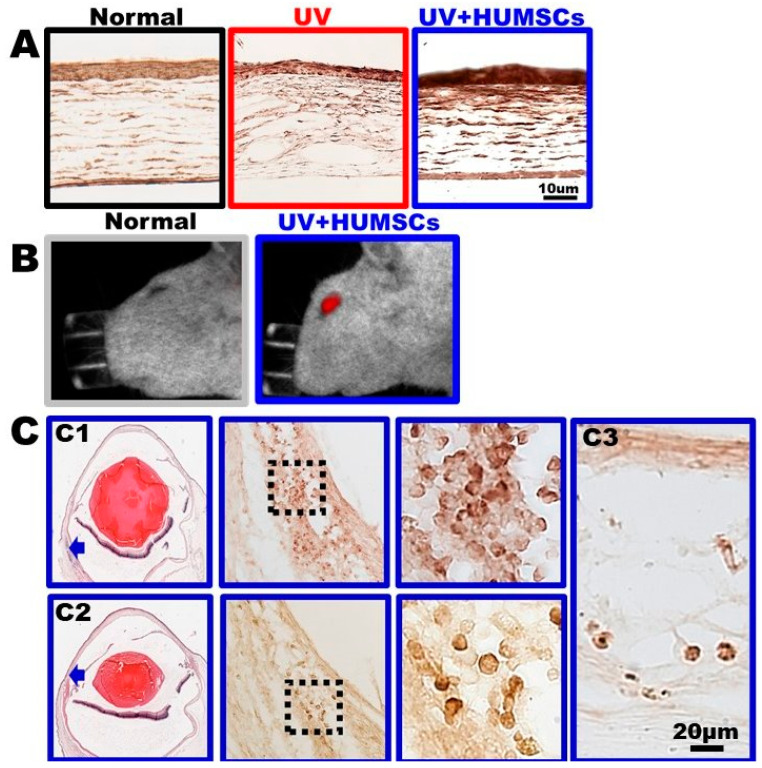
HUMSCs survive and stimulate the expression of ALDH3A1 after UV irradiation. HUMSCs were transplanted into the subconjunctival space of rats after UV irradiation. At 21 days after transplantation, the increased expression of ALDH3A1 was detected in all corneal tissues of the UV + HUMSC group by immunohistochemical staining (**A**). The images show strong red signals after the injection of DiR-HUMSCs in the in vivo fluorescence images (**B**). Sections were stained with hematoxylin and antibody against human nuclei antibody to label the HUMSCs. From low to high magnification, numerous HUMSCs were scattered in the subconjunctival space (arrow) of the UV + HUMSC group (**C** (**C1**,**C2**)), and a small number of HUMSCs were observed in the stroma of the cornea (**C3**).

## Data Availability

The data supporting the reported results were generated during the study and are not publicly available. A summary of the results related to this study can be accommodated upon request from the corresponding author.

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
