# Peer review of "Human Umbilical Mesenchymal Stem Cell Xenografts Repair UV-Induced Photokeratitis in a Rat Model"

_biomedicines, 2022, doi:10.3390/biomedicines10051125_

Round 1
Reviewer 1 Report
This manuscript describes the new treatment based on stem cell base xenograft to treat photokeratitis to prevent loss of vision. The effectiveness is demonstrated in rat model. The stem cell implant improved the healing and reduced the abnormality.
The English of the manuscript can be improved.
The results are supported by clear figures and molecular analysis.
Conclusion must be improved and expanded.
Author Response
Reviewer #1
Comments and Suggestions for Authors
This manuscript describes the new treatment based on stem cell base xenograft to treat photokeratitis to prevent loss of vision. The effectiveness is demonstrated in rat model. The stem cell implant improved the healing and reduced the abnormality.
- The English of the manuscript can be improved.
Thanks for your kindly reminder. We have corrected those in the revised manuscript.
- The results are supported by clear figures and molecular analysis.
Thank you for your suggestion.
- Conclusion must be improved and expanded.
Thanks for your kindly reminder. We have improved and expanded in the revised manuscript.
Reviewer 2 Report
The manuscript “Human umbilical Mesenchymal stem cell Xenografts repair UV-induced Photokeratitis in a rat model” evaluates the therapeutic effect of human umbilical mesenchymal stem cells in UV light corneal injuries rat model.
This study is well designed .
However, a few points should be taken into account:
- Materials and methods
2.1 Isolation and culture of human Mesenchymal stem cells:
- Can the authors give a more detailed protocol for the culture of the cells?
2.2. Experimental animals:
- The rats were treated for how many days? And refer to the figure 1.
- Please provide the number of animals used in each group?
2.3. Experimental grouping:
- Please specify if both eyes were treated or only one eye.
- How many cells were injected?
Figure 1. In B. pictures showing the cornea were too small, can the authors provide larger images.
- Results:
3.1. Ultraviolet light exposure causing corneal injury and corneal opacity
- Authors can use the opacity score to evaluate the transparency of the cornea.
3.4. HUMSC Xenografting repairs UV-Induced corneal epithelial injury
- …. Resulting in incomplete epithelial cells. Can be changed to “incomplete epithelial cell layer”
Again, all images in the figures are too small, it is better to provide more visible and clear images.
- Reviewer don’t see anything in Supplemental figure 3, please provide larger images.
- Results could be completed by adding the analysis of myofibroblastes and MMPs profile in each group.
- Conclusion could be improved.

Author Response
Reviewer #2
- Materials and methods
2.1 Isolation and culture of human Mesenchymal stem cells:
- Can the authors give a more detailed protocol for the culture of the cells?
Thank you for the correction. The mesenchymal tissue in Wharton’s jelly was then diced into cubes 0.5 cm on each side and centrifuged at 250 g for 5 min. After removal of the supernatant fraction, the precipitate that contained mesenchymal tissue was washed with serum-free Dulbecco’s Modified Eagle’s Medium (DMEM) and centrifuged at 250 g for 5 min. After aspiration of the supernatant fraction, mesenchymal tissue in the precipitate was treated with 0.0125% type I collagenase solution (Sigma-Aldrich, St. Louis, MO, USA) at 37 °C for 18 h, washed, and further digested with 0.025% trypsin at 37 °C for 30 min. The treated umbilical tissue then was cultured until the HUMSCs migration and proliferation. Finally, HUMSCs were stored in liquid nitrogen for later transplantation. HUMSCs were collected between the 10th and 15th passages for transplantation into rats in this study. We have corrected those in the revised manuscript.
2.2. Experimental animals:
- The rats were treated for how many days? And refer to the figure 1.
- Please provide the number of animals used in each group.
Thanks for your kindly reminder.
- The light-induced corneal injury experiment was conducted for five consecutive days using a lamp (model UVM-18, P/N 180063-01, UVP Inc., San Gabriel, CA) with a power of 8 watts, a UV wavelength of 302 nm, and an energy of 550 µW/cm2 per day.
- The rats were divided into three groups: Normal group (n=13), UV group (n= 32), and UV+HUMSCs group (n= 13).
We have corrected those in the revised manuscript.
2.3. Experimental grouping:
- Please specify if both eyes were treated or only one eye.
- How many cells were injected?
Figure 1. In B. pictures showing the cornea were too small, can the authors provide larger images.
Thanks for your kindly reminder.
- Only left eyes of experimental animals were treated.
- 2 × 106 HUMSCs were implanted into the subconjunctival space of the lateral sclera using a Hamilton syringe.
We have corrected those in the revised manuscript.
- Results:
2.1. Ultraviolet light exposure causing corneal injury and corneal opacity
- Authors can use the opacity score to evaluate the transparency of the cornea.
On the first day of UV irradiation, there was no significant difference between the appearance of the cornea in the UV group compared with the Normal group, in which clear pupils were observed in both. Corneal transparency did not recover on the 21st day after irradiation, and there were blisters on the corneal surface and abnormal neovascularization in the center of the cornea. In the UV+HUMSC implantation group, on the 14th day after the end of UV irradiation, the corneas of the experimental animals were the end in the UV+HUMSC group were clearer compared with those of the UV group. Otherwise, observation of changes in pathological corneal thickness in In Vivo Images were performed by quantitative analysis of 3D OCT images.
3.4. HUMSC Xenografting repairs UV-Induced corneal epithelial injury
- …. Resulting in incomplete epithelial cells. Can be changed to “incomplete epithelial cell layer”
Thank you for the correction. We have corrected those in the revised manuscript.
Again, all images in the figures are too small, it is better to provide more visible and clear images.
- Reviewer don’t see anything in Supplemental figure 3, please provide larger images.
- Results could be completed by adding the analysis of myofibroblastes and MMPs profile in each group.
- Conclusion could be improved.
- Thanks for your kindly reminder. We have changed images in Supplemental figure 3.
- Previous histopathological analysis to examine the change in inflammatory cell density, and blood vessel counts, along with the expression profiles of matrix metalloproteinase and vascular endothelial growth factor. In our study, keratocytes were stained with anti-ALDH3A1 to assess keratocyte distribution in the cornea and identify the distribution of myofibroblasts associated with fibrosis in the cornea.
- Thank you for the correction. We have corrected those in the revised manuscript.
